



# Estimating karst groundwater recharge from soil moisture observations - A new method tested at the Swabian Alb, Southwest Germany

Romane Berthelin[1], Tunde Olarinoye[1], Michael Rinderer[1], Matías Mudarra[2], Dominic Demand[3], Mirjam Scheller[1], Andreas Hartmann[4,1]

[1]Chair of Hydrological Modeling and Water Resources, Freiburg University, Freiburg, 79098, Germany

[2]Department of Geology and Center of Hydrogeology of the University of Málaga, Faculty of Science, E-29071, Málaga, Spain

[3]Chair of Hydrology, Freiburg University, Freiburg, 79098, Germany

[4]Institute of Groundwater Management, Technical University of Dresden, 01069 Dresden, Germany

*Correspondence to*: Romane Berthelin (romane.berthelin@hydmod.uni-freiburg.de)

## Abstract

Understanding groundwater recharge processes is important for sustainable water resource management. Experimental approaches to study recharge in karst areas often focus on analysing the aquifer response using a disintegration of its outlet signals, but only a few directly investigate the recharge processes that occur at the surface of the system. Soil moisture measurements have a high potential to investigate water infiltration to deeper soil depth or epikarst with an easy and not too intrusive installation. They can yield long-term measurements with high temporal resolution. Using these advantages, we developed and tested a method to estimate recharge based on soil moisture measurements. The method consists of the extraction of linked events in rainfall-, soil moisture and discharge time series and a subsequent fitting of the parameters of a simple drainage model to calculate karst recharge from soil moisture metrics of individual events. The fitted parameters could be interpreted in physically meaningful terms and were related to the properties of the karstic system. The model was tested and validated in a karst catchment located in Southwest Germany with hourly precipitation, soil moisture, and discharge data of eight years duration. The soil moisture measurements were distributed among grassland (n = 8) and woodland areas (n = 7) at 20 cm depth. A threshold of about 35 % (±8 %) of volumetric water content was necessary to initiate effective infiltration. Soil moisture averaged during the wetting period of each event was the best metric for the prediction of recharge. The model performed reasonably well estimating recharge during single rainfall events. It was also capable to simulate 88 % of the average annual recharge volume despite considerable differences in the performance between years. The event-based approach is potentially applicable to other karstic systems where soil moisture and precipitation measurements are available to predict karst groundwater recharge.





## 1 Introduction

Karst aquifers provide a significant portion of the resources used for drinking water supply in many regions of the world. In regions where alternative sources of drinking water exist, karst water resources are often avoided due to unstable discharge

regimes and high vulnerability to pollution (Stevanović, 2019). In other regions where no alternatives exist, pressures on karst groundwater resources increase, and this raises the need for improving protective measures and water management to avoid the depletion of carbonate aquifers (Xu et al., 2018), especially under climate change context. Groundwater recharge process understanding is therefore important for sustainable water resource management and governance. Experimental methods to evaluate recharge in karst areas often focus on analysing the aquifer response using a disintegration of its outlet

signals measured at the karstic spring (Goldscheider and Drew, 2007). Discharge and physiochemical measurements, as well as natural or artificial tracers, can provide a lot of information concerning the hydrological functioning of the system, and consequently on its recharge processes. For example, spring hydrographs and hydrochemical signal monitoring bring information about the subsurface structure and dynamics of karst aquifers (Jeannin et al. 2007; Mudarra & Andreo, 2011; Perrin et al., 2003). Natural tracers such as water isotopes can be used to understand the transit times and dispersion of water

entering the entire catchment (Aquilina et al., 2005; Maloszewski et al., 2002), and to help for estimating mean recharge altitude of each sub-catchment (Sappa et al., 2018). In addition, dye tracer tests are a powerful tool to investigate flow paths and times through karst systems (Goldscheider et al., 2008). Furthermore, modeling applied to predict recharge is usually also evaluated with observations at the system outlet (Chen et al., 2017; Mudarra et al., 2019; Ollivier et al., 2020). In other cases, GIS-based methods that use spatial information about geology, soil types, vegetation, mean annual precipitation, etc.,

are often used to derive time-averaged spatial distribution of karst recharge (Andreo et al., 2008; Radulovic et al., 2011; Allocca et al., 2014). Other experimental methods conducted at the surface of karst systems, such as geophysical approaches, allow the investigation of the edaphic and hydraulic properties with influence on recharge mechanisms. For example, different geophysics methods were used to highlight the role of the porous rock matrix that can act as a seasonally varying storage in the unsaturated zone (Carrière et al., 2016). However, most of these methods are time-consuming and/or

expensive to apply.

Despite the important role of the surface heterogeneity and its processes on recharge (Williams, 2008), this heterogeneity makes it difficult to assess and predict groundwater recharge from the earth's surface. Although progress in the understanding of subsurface heterogeneity in karst media has been made in the last years, few studies have yet applied experimental approaches to characterize karst recharge mechanisms with observations collected directly at the shallow

subsurface. This includes the soil and the epikarst, which is the superficial weathered rock. Tobin et al. (2021) developed a conceptual model of the hydrological processes occurring at two different epikarst zones based on the study of its hydraulic and hydrochemical responses to different storm events. Precipitation amount, intensity, and seasonality were the main factors impacting the outflow response for both independent sites in this study. However, they also mention that soil and vegetation



have an important influence on recharge mechanisms and advice further investigation of subsurface processes to understand
their effect on the aquifer as a whole.

Various subsurface flow pathways and subsequent groundwater recharge were found to be depending on changes in shallow
soil moisture conditions (Perrin et al. 2003). Using lysimeters to analyse the hydrochemical signal of water from the soil, an
influence of preferential flow pathways in soil on karst recharge processes was confirmed (Tooth & Fairchild 2003). Ries et
al. (2015) measured shallow soil moisture at a Mediterranean karst site and used this data in a model to simulate percolation
towards the saturated zone. They concluded that simulated fluxes from a plot scale measurement is not directly transferable
to a larger scale, but it may help to understand processes influencing temporal and spatial groundwater recharge such as the
fast infiltration of water during heavy precipitation events. In other studies, soil moisture data and simulation tools were
jointly applied in order to assess recharge in karst terrains (Sarrazin et al., 2018; Ireson and Butler, 2011), but similar to Ries
et al. (2015), transferability of results to larger scales such as the entire karst system remained uncertain due to (1) a low
number of locations where soil moisture was observed and (2) the lack of evaluation with independent recharge observations
at the aquifer scale. Messerschmid et al. (2020) did manage to simulate recharge coefficient only based on limited locations
of soil moisture observation and on ungauged Mediterranean karst basin. This was possible because of long-term
observations and well-chosen representative locations for specific formations allowing their transferability to comparable
catchments.

In other non-karstic geological settings, soil moisture measurements conducted at a high temporal resolution have been used
in a few studies to investigate infiltration related processes (Demand et al., 2019; Martini et al., 2015). Schaffitel et al. (2021)
developed a data-driven water-balancing framework to derive water fluxes from meteorological data and soil moisture
measurements. One of the steps of this framework was the calculation of the soil water balance, which included the fitting of
a drainage model that could be used to predict drainage from soil moisture measurements. The promising results of these
studies allow assuming that shallow soil moisture measurements might be informative for estimating subsurface flow and
groundwater recharge in karst systems. In fact, as infiltration and recharge are less delayed in karst aquifers compared to
other geological settings, one could assume to find an even stronger relation between soil moisture and karst spring
discharge.

Therefore, in this study, we developed and tested a new methodology to estimate karst aquifers recharge from shallow soil
moisture measurements. In particular, we (1) extracted and attributed precipitation events, soil moisture events, and recharge
events to each other, (2) conducted statistical analyses to study the relationship between soil moisture and recharge and
applied a drainage model to simulate recharge. The parameters of the empirical relations that we derived can be interpreted
in physical meaningful terms and are useful to characterise karst system properties. As groundwater recharge mainly takes
place during and shortly after rainfall events we follow an event-based approach. For a proof of concept, we applied the
method to a collection of soil moisture measurements and discharge over eight years at the karstified region of Swabian Alb
in Southwest Germany.





## 2 Methods and data

In order to investigate the link between soil moisture and groundwater recharge, an event scale approach was applied. First, precipitation events, soil moisture events, and recharge events were identified and extracted subsequently from continuous time series of precipitation, soil moisture and discharge. In a second step, the precipitation events were attributed to the correspondent soil moisture events and to the correspondent recharge events while accounting for temporal delays between the three-time series with a simple temporal buffer. That way, our event scale approach avoided complexity of considering the time scales of water movement through the soil and groundwater aquifer and therefore allows to focus on the volumetric relationships of rainfall, soil moisture, and recharge.

Combinations of precipitation events that produced one clearly identifiable and causally linked soil moisture event and one recharge event were selected for statistical analysis to study the relationship between different soil moisture and recharge metrics such as the average soil moisture during an event and the recharge volume. In addition, the parameters of a drainage model, based on the unit gradient approach (Yeh, 1989; Hillel, 1998), was fitted to the data to describe the relation between soil moisture and groundwater recharge. Finally, the drainage model was evaluated by calculating recharge volumes for all soil moisture events over the eight years of the study period and comparing them to recharge volumes inferred from discharge measurements. Our approach is exemplified with an experimental dataset collected at the Swabian Alb in southwest Germany.

### 2.1 Event selection

#### 2.1.1 Precipitation, soil moisture and recharge events selection

In a first step, precipitation, soil moisture and recharge events were extracted based on different thresholds from the observed precipitation, soil moisture and stream discharge time series independently. The event selection criteria were as follows: (1) Similar to Demand et al. (2019), a precipitation event was defined to have at least 1mm of total rainfall. Rainfall events were separated, if there was at least 24 hours of no rainfall between the events (Fig.1a). (2) The start of a soil moisture event was defined as an increase in volumetric water content of at least 1 %, which corresponds to the accuracy of the probes. The end of the event was set to the start of the following event (Fig. 1b).  (3) We use discharge as a proxy for groundwater recharge at the event time scale (see elaboration below). The start of a recharge event was defined by the time when the three-day running average of the observed slope of the discharge time series changed from negative to positive values. Similar to the selection of soil moisture events, the end of the recharge event was considered as the start of the following recharge event (Fig. 1c).

From the selected rainfall events, different metrics were extracted: the total volume of rainfall, the duration of the event, and its mean intensity. From the selected soil moisture events, antecedent soil moisture (which is defined as the volumetric water content at the time of the start of the soil moisture response), soil moisture maximum, soil moisture response amplitude, and mean soil moisture during the event were extracted. In addition, the so-called "wetting period" was defined as the time


between the start of the soil moisture response and the end of the precipitation event, which is the time during which one
expects groundwater recharge. For this period, the mean soil moisture during the wetting period was extracted. From the
selected recharge events, the volume of recharge and the recharge rate was derived. The volume of recharge during each
event was estimated assuming that the stream discharge of a karst spring is a good proxy for recharge. This is plausible as
karst systems are highly responsive to precipitation. Due to preferential pathways, water transfer through the vadose zone is

usually quick (Hartmann et al., 2021) and surface runoff is usually marginal (Hartmann et al., 2012; Worthington et al.,
2016). In addition, an event scale approach allows the evaporation to be assumed low as the observation time is limited. For
these reasons, it is assumed that all water that does not evaporate or remains in the soil, will contribute to recharge. A simple
approach based on available groundwater storage at the beginning and the end of each event was used to estimate the per-
event discharge volume. Firstly, the recession events were extracted from the discharge time series using the extraction

procedure of (Vogel and Kroll, 1996), after which the recession constant, $k$, was estimated using a linear storage-discharge
function. Then, the total volume, $V_T$, of water for each event cycle (from the first positive slope change to the next) was
calculated using the integral area approach. The volume of groundwater, $V_0$, stored at the beginning of the event was
calculated by dividing the discharge, $Q_0$, corresponding to the start of the event by the recession constant, $k$. If we assume
there is no recharge event, the theoretical decrease of $Q_0$ till the end of the event cycle can be estimated by linear

extrapolation. This way, the theoretical discharge, $Q_{th}$ that would be reached at the end of the event cycle, the groundwater
volume, $V_i$, that would be discharged as well as the volume, $V_{th}$, that would be stored were all calculated. The change in
groundwater storage was estimated by the difference between volume, $V_0$, stored at the beginning of the event, and the
theoretical volume, $V_{th}$, stored at the end. The event recharge is then given by the difference between the total volume, $V_T$,
theoretical discharge volume, $V_i$, and change in groundwater storage. The total volume of recharge divided by the wetting

period yielded the recharge produced during the event. The recharge rate corresponded to recharge volume divided by the
precipitation volume.

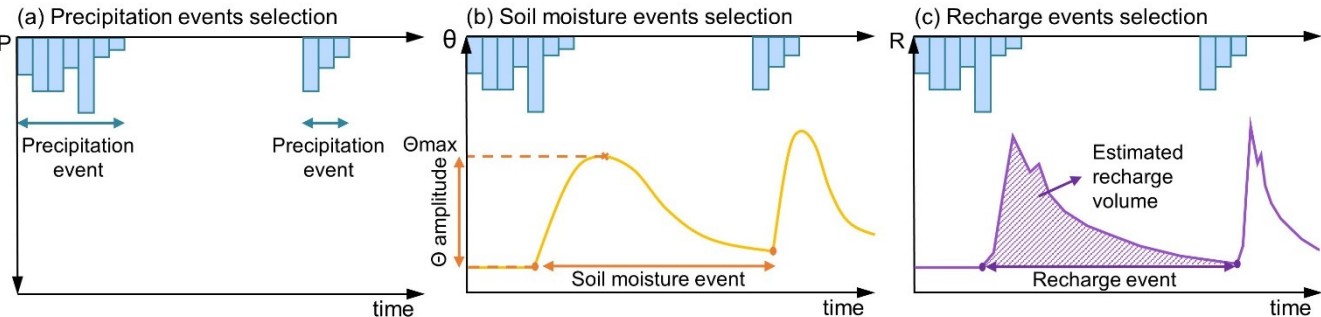

**Figure 1: Event selection method applied in this study exemplified for two precipitation (a), soil moisture (b) and recharge (c)
events.**

**2.1.2 Attribution of precipitation, soil moisture and recharge events to each other**

As rainfall, soil moisture and discharge typically respond delayed relative to each other, a procedure to link them was





necessary. Not all precipitation events initiate a soil moisture event or a recharge event. For that reason, we attributed a precipitation event to a soil moisture event when it occurred within the period between the start, and the peak of a soil moisture event. To account for natural delays between precipitation and soil moisture we allowed for an additional temporal

buffer, which was determined by the cross-correlation between the precipitation and the soil moisture time series (for details see Delbart et al., 2014). If the precipitation event happened during the recession of the soil moisture event it was counted as a precipitation event that did not produce a soil moisture response (Fig. 2a). The same approach was applied to link precipitation event with recharge events (Fig. 2b), and soil moisture events with recharge events (Fig. 2c). If a soil moisture or a recharge event was not linked to a precipitation event, it was excluded from the analysis based on the assumption that a

soil moisture or recharge event cannot be produced without precipitation.

The number of events selected was counted for each soil profile, as well as events occurring on the entire catchment, which were counted as one event is an event happening at least one time at, at least one location.

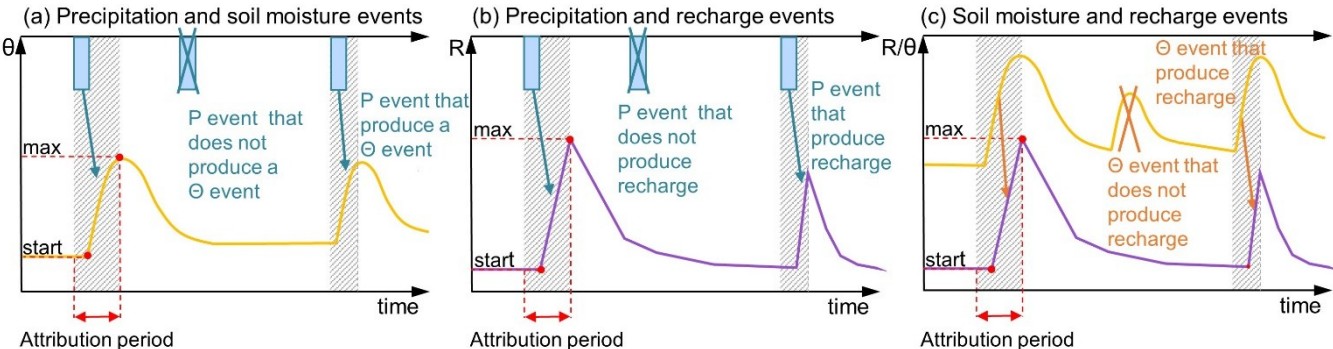

**Figure 2: Event attribution method applied in this study exemplified for the attribution of precipitation events to soil moisture**
**events (a), for the attribution of precipitation events to recharge events (b), and for the attribution of soil moisture events to**
**recharge events (c). (P: precipitation, R: recharge, Θ: soil moisture).**

## 2.2 Quantitative linkages between soil moisture characteristics and recharge

### 2.2.1 Correlation analysis between precipitation, soil moisture and discharge

In order to find a link between soil moisture and recharge, the correlation between different soil moisture metrics and
recharge volumes was calculated. As soil moisture metrics, we used the soil moisture value at the beginning of the event (antecedent soil moisture), the maximum value of soil moisture reached during the event (soil moisture peak), the median value of soil moisture during the event (soil moisture median), the soil moisture response amplitude (soil moisture amplitude), the averaged soil moisture during the event (soil moisture mean), and the average value of soil moisture during the wetting period (mean soil moisture during wet). We assume that these metrics capture important aspects of the soil
hydrological dynamics. Only combinations of at least one rainfall event causing one soil moisture event and one discharge





event were considered in the following statistical analysis and for the model calibration. This selection of single causal events allowed a more reliable calibration by avoiding the potential interferences of multiple soil moisture events on recharge. The relation between these different soil moisture metrics and recharge volume, as well as the recharge rate was explored. In order to detect non-linear relationships, a Spearman rank correlation was applied. The closer the Spearman's

rank correlation coefficient ($\rho$) is to +1 or -1 is the stronger is the relation between the two tested variables. Its significance is evaluated using the probability value (p- value). In this study, we considered the results statistically significantly correlated if p-value < 0.01.

**2.2.2 Drainage model based on the unit gradient approach**

In addition to the correlation analysis, we used a more physically-oriented approach to describe the relation between soil
moisture and recharge. We fitted the unit gradient model (Hillel, 1998) with its two parameters ($B$ and $k_S$) (Schaffitel et al., 2021). Vertical water flow, considered here as a proxy for groundwater recharge, is defined as:

$$(1) \qquad Q = k_s \cdot \left( \frac{\varTheta - \varTheta_r}{\varTheta_s - \varTheta_r} \right)^{\frac{2+3B}{B}}$$

Here, $k_s$ [mm d$^{-1}$] and $B$ [-] are calibration parameters representing saturated hydraulic conductivity and the pore size distribution index, respectively. $\varTheta r$ [vol %] and $\varTheta s$ [vol %] correspond to the residual water content and the saturated water content. In this study, the soil moisture minimum and maximum measured during soil moisture time series were used, as in Schaffitel et al., (2021). The relationship is based on the Burdine-Brooks-Corey parametrization of hydraulic conductivity (Brooks and Corey, 1964). The soil water storage term in the water balance equation is substituted by a soil moisture term.
By this, the vertical water flow, or groundwater recharge $Q$ [mm d$^{-1}$], can be expressed as a function of soil moisture that is similar to soil water storage. Multiplied by the duration of the event, we can obtain the recharge volume [mm] for each individual event.

To calibrate the model, a Monte Carlo approach was applied. The $k_s$ parameter was sampled between 0 and 50 [mm d$^{-1}$] with
a step resolution of 0.1 [mm d$^{-1}$]. These ranges correspond to the hydraulic conductivity for clayey and silt-sandy soils. The dimensionless parameter $B$, which represents the pore size distribution, was sampled between 0 and 5 with a step resolution of 0.05. The best model parameters were selected by minimizing the root mean square error (RMSE) of the model. Uncertainty in identifying the best parameters was accounted for by selecting also the 10 % best simulations (with the 10 % lowest RMSE).


For validation of the model, annual recharge was calculated using the best overall parameter set and compared to the





observed annual recharge volume. To compute the annual recharge using the drainage model, all soil moisture events detected with the selection method were kept for each hydrological year (beginning of October - end of September). For each soil moisture event, the groundwater recharge $Q$ [mm d$^{-1}$] was computed using the mean soil moisture during the wetting

period ($\Theta$) which is the best soil metric identified during the correlation analysis described in the Sect. 2.2.1 (see results in table 2). The result obtained for each event was then multiplied by the corresponding wetting period, in order to get the recharge volume [mm] for each event. Finally, annual modelled recharge estimates were compared to the annual observations of recharge obtained as explained in Sect. 2.1.1. Acceptable agreement would indicate that the event-based method could be applied to longer time series of soil moisture measurements to predict karst groundwater recharge.

**2.3 Study site and data description**

To exemplify the applicability of our approach, we used an experimental dataset collected within in the catchment of the Große Lauter River in the Swabian Alb, Southwest Germany (Fig. 3) provided by the Biodiversity Exploratory research project (Table 1). According to the description made by Goldscheider (2005), the geology in the Swabian Alb is composed of 300 to 400 m thick karstified carbonate rocks from the Upper Jurassic. This formation is covered in parts by Molasse

sediments and glacial deposits. The soil is shallow (25 to 32 cm) with a silty clay texture (Gimbel et al., 2016). The Große Lauter surface catchment size is 325 km² with an altitude between 504 and 896 m above sea level. Using long-term estimates of water balance components from the Water & Soil Atlas of the state of Baden Wuerttemberg (WaBoA (Ed.), 2012), we estimated the size of the subsurface catchment to be 170 km². Assuming that there is no surface runoff due to the karstic properties of the system, we use the size of the subsurface catchment for our further analysis.


Discharge data of the Große Lauter river was available daily and hourly. The mean annual discharge of the river is 1.38 m³ s$^{-1}$ with a minimum of 0.45 m³ s$^{-1}$ and a maximum of 3.48 m³ s$^{-1}$ for the control period (from Nov. 2009 to Sep. 2017). Mean annual precipitation and mean annual air temperature of the site are 940 mm and 6.5°C, respectively (Gimbel et al., 2016), with some snowfall during the winter season. In total, four climate stations are located in the studied catchment, which

measure precipitation on an hourly time interval. Thiessen polygons were used to compute an interpolated precipitation dataset for the catchment. Snowmelt was considered using a degree-day approach (Lindström et al., 1997). A detailed description of the routine is provided in Parajka et al. (2007). The snow melt parameters were adapted from Schulla (1997) and Hartmann et al. (2013) who applied the same routine at nearby sites. The hourly sum of liquid precipitation and snowmelt was used as input to the karst system in the following analyses.


The soil moisture measurements were collected with Decagon 5TM probes (Frequency Domain Reflectrometry) installed at 20 cm depth. They were used to measure soil water content at an hourly resolution. For our analysis, we used 15 soil moisture measurement locations covering a period of eight years from 2009 to 2017 that were distributed between the two types of vegetation: seven in woodland and eight in grassland (Fig. 3). The woodland areas on the catchment represent 58 %





of the cover, while the open areas correspond to 42 %.

Two spatially averaged soil moisture time series were calculated, one from all grassland (G) time series and one from all woodland (W) time series. These two-time series reflect the average soil moisture dynamics of the grassland and the woodland sites in the catchment and are less affected by sites specific heterogeneity than the time series of individual sites.

In a same way, a catchment-average soil moisture time series was calculated as the average of the grassland and woodland time series weighted by the percentage of land cover of the catchment. This combined time series reflects the catchment-average soil moisture dynamics. These three average time series were used for the analysis in this study. In addition, a time series of standard deviations for each average time series was computed to quantify the spatial variability of soil moisture measurements among the profiles in grassland, woodland areas, and over the entire catchment at each time step.


**Table 1: Data set description.**

| Data | Unit | Temporal resolution | Time period | Data gap | Source |
|---|---|---|---|---|---|
| Precipitation | mm | hourly | | 3.5 % | Biodiversity Exploratory research project (DFG Priority Programme 1374) – Core Project Instrumentation |
| Soil moisture | vol % | hourly | From Nov. 2009 to Sep. 2017 | 12 % | |
| Discharge | $m^3\ h^{-1}$ | hourly & daily | | 0 % | Environment Agency of the German state of Baden-Württemberg (LUBW) |





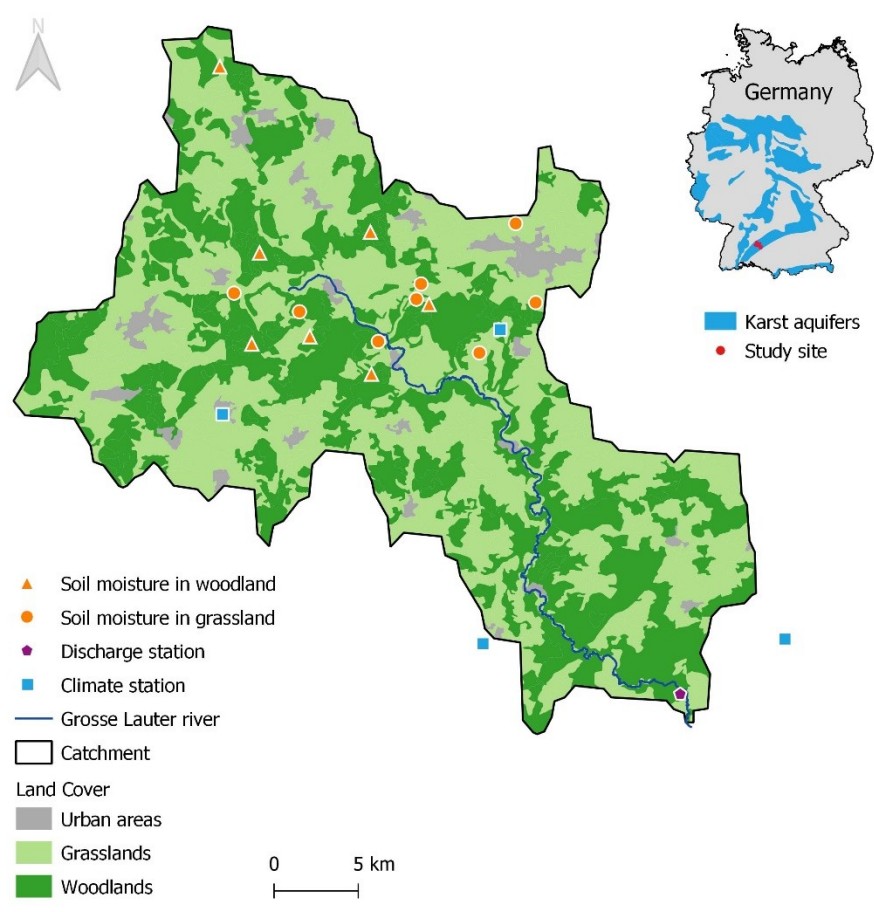

**Figure 3: Map of the Große Lauter catchment with the location of the soil moisture profiles, the discharge gauging station and the four climate stations. (Land cover: Corine Land Cover CLC, (2018) modified. Karst aquifers: Chen et al. (2017)).**

## 3 Results

### 3.1 Event selection

Events were extracted from the precipitation, soil moisture, and discharge time series. In total 455 precipitation events were
identified, and 266 soil moisture responses from of at least one of the soil moisture probes on the catchment. For the analysis, 97 soil moisture events from the grassland time series, 143 soil moisture events from the woodland time series, and 190 recharge events were selected (Fig. 4a). On average per year, the number of precipitation events was about 50. The number of recharge events per year was about 20. On the entire catchment, the number of soil moisture events was about 30 per year. In grassland, the average number of soil moisture events per year was slightly above 10, while in woodland, the





number was about 15. The heterogeneities in the number of events detected for each profile were also higher in woodland than in grassland.

The percentage of precipitation events that caused both soil moisture and recharge events was 14 % in grassland and 19 % in woodland (Fig. 4b). The percentage of precipitation events that caused soil moisture events but not recharge events was 36 % for grassland and 35 % for woodland. In total, a higher percentage of precipitation events that caused any type of soil

moisture response was found in woodland compared to grassland. The fraction of precipitation events that were attributed to recharge but not linked to a soil moisture response was 33 % in grassland and 28 % in woodland. 17% and 18% of all precipitation events in grassland and woodland caused neither soil moisture nor recharge to respond.

In order to study the link between soil moisture and recharge, only the combinations of precipitation events that were attributed to one soil moisture event and also one recharge event were used for the following empirical analysis and for the

model calibration procedure.  For calculating the recharge volume over the eight-year study period, all events were used.

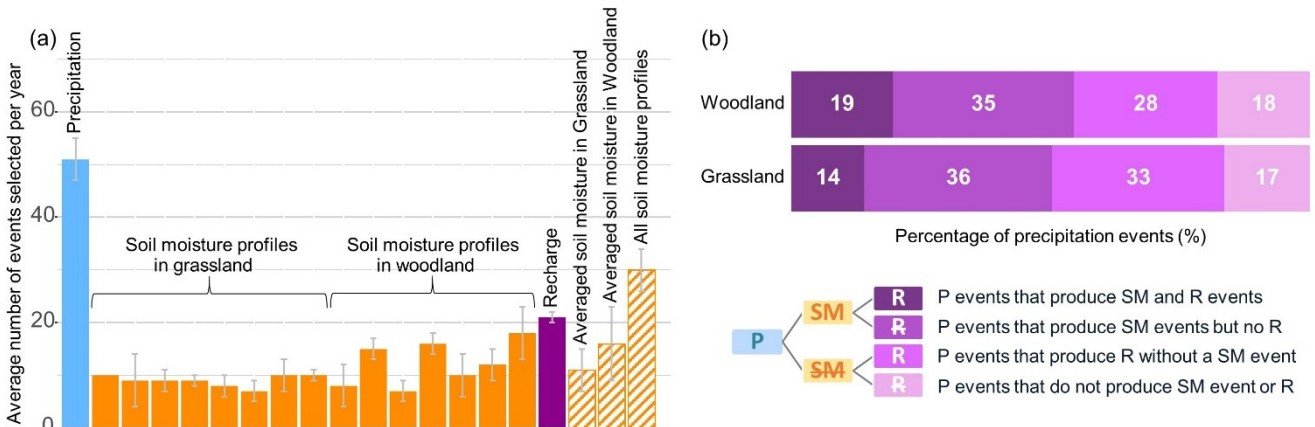

**Figure 4: a. Averages number of precipitation (P) events, soil moisture (SM) events, and recharge (R) events selected for each hydrological year. Each orange bar corresponds to one soil profile. b. Percentage of precipitation events attributed to soil moisture and recharge events with the attribution method (number total of precipitation events = 455).**


### 3.2 Correlation analysis between soil moisture and discharge

For those rainfall events that produced both a single soil moisture and a recharge response, spearman rank correlations ($\rho$) between soil moisture metrics and recharge metrics (recharge volume and recharge rate) of the corresponding events were calculated. Results are shown in table 2. The highest correlation coefficients were found between the recharge volume and

soil moisture metrics from grassland areas ($\rho$ up to 0.74) while those for woodland areas ($\rho$ up to 0.69) and the combined areas  ($\rho$ up to 0.67) were slightly smaller but still significant. From all soil moisture metrics, mean soil moisture during the wetting period showed the highest correlation with the recharge volume for all three groups (grassland, forest and combined areas) but also the soil moisture mean and median yielded comparably high correlations. Only the metric soil moisture





amplitude was not significantly correlated with recharge volume and recharge rate. The soil moisture amplitude showed a
low correlation with the recharge volume and with the recharge rate, for all grassland, woodland and the combined areas. All
other soil moisture metrics were statistically significantly correlated ($\rho > 0.5$, p value $< 0.01$) with recharge volume and
recharge rate with slightly higher correlation coefficients for recharge volume than recharge rate.

**Table 2: Spearman rank correlation ($\rho$) between soil moisture (SM) metrics and recharge descriptors for grassland, woodland and the combined areas. (Colours red: $\rho < 0.2$, orange: $\rho < 0.4$, yellow $\rho < 0.6$, green $\rho < 0.8$).**

| | | SM start | | SM peak | | SM median | | SM amplitude | | Mean SM | | Mean SM during wetting | |
|---|---|---|---|---|---|---|---|---|---|---|---|---|---|
| | | $\rho$ | p value | $\rho$ | p value | $\rho$ | p value | $\rho$ | p value | $\rho$ | p value | $\rho$ | p value |
| Grassland | Recharge volume | 0.67 | < 0.01 | 0.73 | < 0.01 | 0.74 | < 0.01 | 0.32 | < 0.1 | 0.73 | < 0.01 | 0.74 | < 0.01 |
| | Recharge rate | 0.6 | < 0.01 | 0.59 | < 0.01 | 0.58 | < 0.01 | 0.12 | > 0.1 | 0.58 | < 0.01 | 0.55 | < 0.01 |
| Woodland | Recharge volume | 0.57 | < 0.01 | 0.59 | < 0.01 | 0.62 | < 0.01 | 0.12 | > 0.1 | 0.62 | < 0.01 | 0.69 | < 0.01 |
| | Recharge rate | 0.6 | < 0.01 | 0.57 | < 0.01 | 0.59 | < 0.01 | -0.02 | > 0.1 | 0.6 | < 0.01 | 0.59 | < 0.01 |
| Combined areas | Recharge volume | 0.54 | < 0.01 | 0.6 | < 0.01 | 0.63 | < 0.01 | 0.38 | < 0.05 | 0.63 | < 0.01 | 0.67 | < 0.01 |
| | Recharge rate | 0.56 | < 0.01 | 0.53 | < 0.01 | 0.53 | < 0.01 | -0.03 | > 0.1 | 0.54 | < 0.01 | 0.52 | < 0.01 |

As mean soil moisture during the wetting period was highest correlated with the recharge volume for all three groups, this
combination was used for the subsequent analyses. Figure 5 shows the relationship between the mean soil moisture during
the wetting period and the recharge volume for each event selected for the grassland, woodland, and the combined areas. The
standard deviation of the mean soil moisture during the wetting period caused by averaging across all grassland, all
woodland, and all monitoring sites of the catchment, is shown using a colour scale. It allows the assessment of the spatial
variability of soil moisture response due to site specificities of individual profiles.

The visual analysis clearly indicated an exponential relationship between the recharge volume and mean soil moisture during
the wetting period. The maximum soil moisture values in the grassland reached almost 45 %, while values remained below
40 % in woodland. It seems that when the soil moisture reached the threshold of about 35 %, the recharge volume started to
increase for all three datasets. In grassland areas, the standard deviation of the mean soil moisture during the wetting period
got lower with increasing soil wetness, especially when exceeding the 35 % threshold. Under these conditions, the soil
moisture measured at the different profiles across the grassland sites was getting more consistent. This is not observed for the
profiles in woodland areas, where the measurements were more disparate. The results using the combined areas of grassland
and woodland sites showed an average behaviour of the one observed in the two respective areas. The exponential shape of
the distribution of the data points was similar to the one for woodland, and the standard deviation of the mean soil moisture
during the wetting period got lower with increasing soil wetness, as observed for grassland but in an attenuated way.





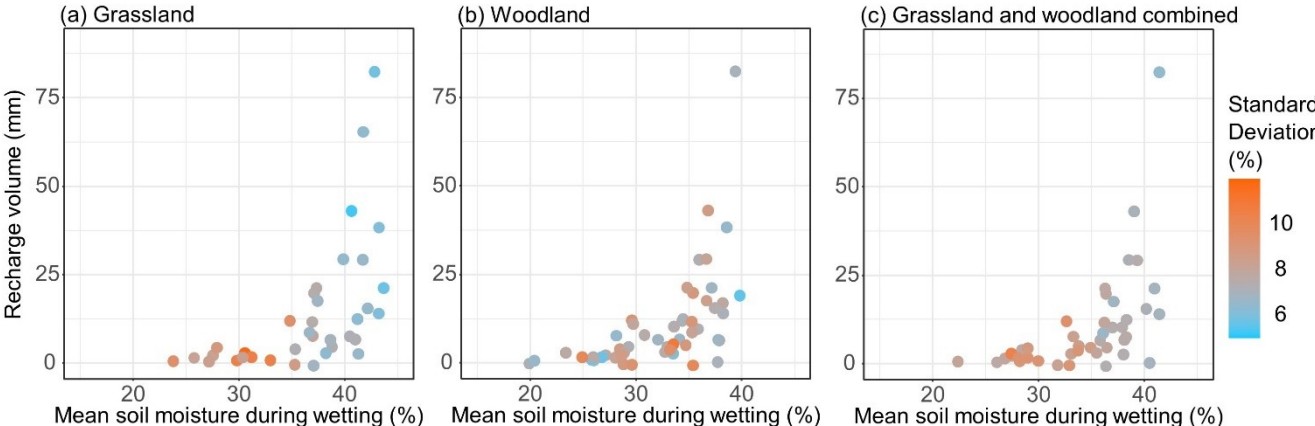

**Figure 5: Relationship between recharge and mean soil moisture during the wetting period. Each data point corresponds to one selected event; the colour indicates the standard deviation of mean soil moisture during the wetting period caused by the variability within the grassland (a), woodland (b) and the combined areas (c).**

### 3.3 Drainage model based on the unit gradient approach

The exponential relationship between the recharge volume and the soil moisture during the wetting period, already indicated that the drainage model was an adequate choice, as it is based on an exponential function. Figure 6 shows the results of the model calibration. The simulated flux using the drainage model was in millimetres per day, it had to be multiplied by the duration of events to obtain the recharge in millimetres as described in Sect. 2.2.2. In grassland, compared to woodland or the combined areas, the model seemed to present more difficulties to simulate the highest values of recharge.

In grassland, the $k_s$ was estimated at 11.4 mm d$^{-1}$and $B$ at 5 for a RMSE of 3.58. In woodland, the $k_s$ was estimated higher at 49.2 mm d$^{-1}$ and $B$ lower at 0.5 for RMSE of 4.72. With the combined areas, the RMSE was 3.82, with the $k_s$ estimated at 34.8 and $B$ at 0.45. The 10 % best simulations (with the 10 % lowest RMSE) were also applied and were represented in Fig. 6. In grassland, the $ks$ was estimated between 8.1 and 16.7 mm d$^{-1}$ with a $B$ between 1 and 5. In woodland, the $ks$ was estimated between 17.2 and 50 mm d$^{-1}$ with a $B$ between 0.40 and 5. For the combined areas, the $ks$ was estimated between 13.9 and 50 mm d$^{-1}$ with a $B$ between 0.25 and 5.





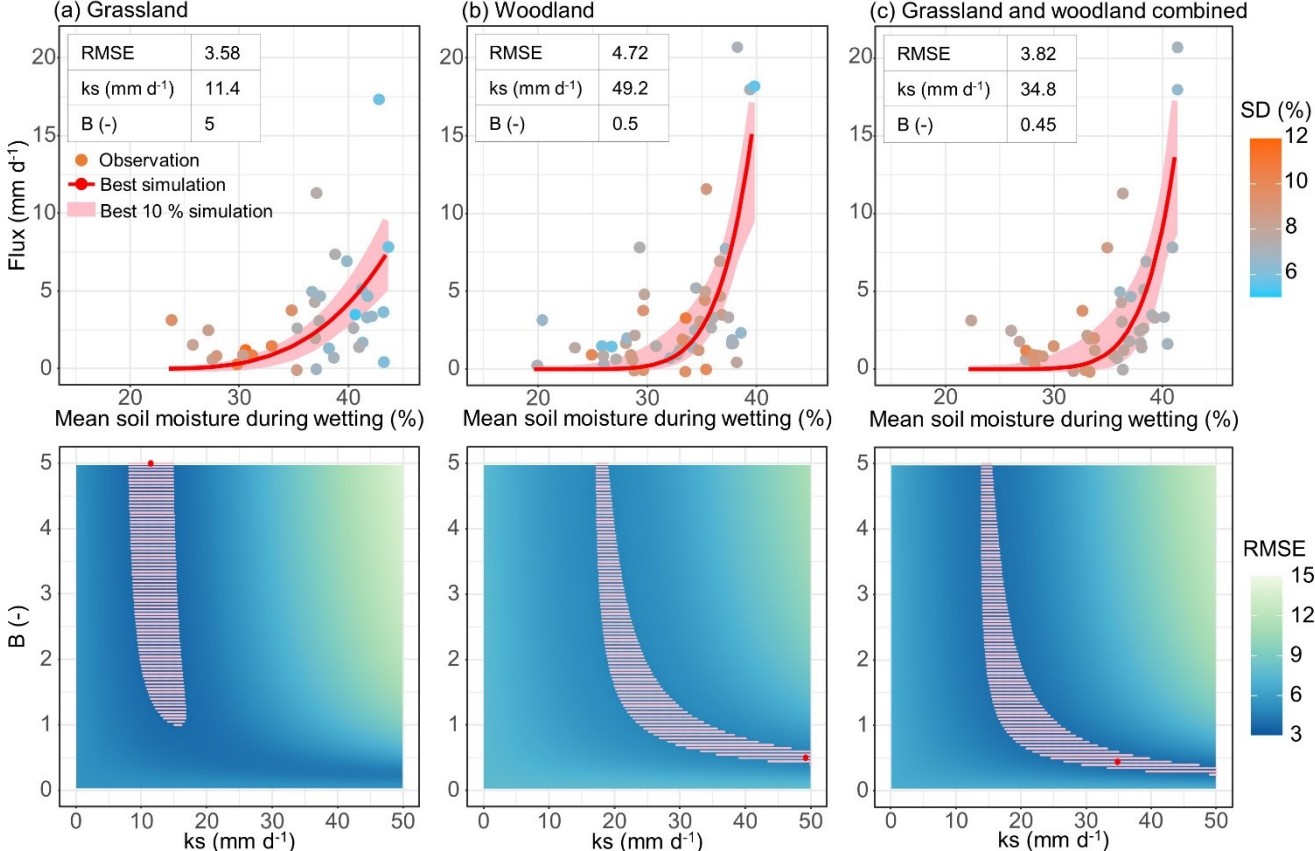

**Figure 6: Results of the Monte-Carlo parameter calibration of the drainage model for grassland (a), woodland (b), and the combined areas (c). The best simulation is the one presenting the lowest RMSE. Upper panels: The observed fluxes as a function of the soil moisture during the wetting period for each event (points) coloured by the standard deviation of mean soil moisture during the wetting period (SD), and the simulated fluxes (red line) using the drainage model with the fitted parameters. Lower panels: The RMSE of the drainage model for different combinations of $ks$ and $B$ parameters tested in the Monte Carlo calibration procedure, the 10 % lowest RMSE (dashed pink), and the $ks$ and $B$ parameters with the lowest RMSE (red point).**

The drainage model with its fitted $ks$ and $B$ parameters was used to estimate recharge volumes for each hydrological year of the study period (table3). All soil moisture events selected for each year and each grassland, woodland, and the combined areas was considered for this analysis. On average, the model was capable to predict 88 % of the observed recharge volume calculated from measurements using the combined areas. This number was about 70 % when calculating the annual recharge volume only based on the grassland or the woodland data. The results were however disparate depending on the studied year. For example, in 2010 the drainage model yielded 119 % and 134 % of the observed annual recharge volume while in 2012, the drainage model yielded between 42 % and 57 % of the annual recharge volume. In 2015 and 2016, the drainage model





yielded between 30 % and 40 % of the observed annual recharge volume, and in 2014 between 218 % and 348 % for the observed annual recharge volume. The results for the recharge rates presented similar patterns. For example, in 2010, the model yielded a recharge rate between 39 and 44 % compared to the observed recharge rate of 33 %. Again, in 2014 the drainage model yielded the largest deviation (between 62 and 98 % of recharge rate) from the observed recharge rate of 28 %. In general, the combined grassland and woodland data lead to better results.


**Table 3: Annual recharge volume and annual recharge rate calculated from measurements and simulated using the fitted drainage model for the grassland, woodland, and combined areas.**

| Year | Recharge volume observed (mm) | Recharge volume modelled - grassland (mm) | Recharge volume modelled - woodland (mm) | Recharge volume modelled - combined (mm) | Recharge rate observed (%) | Recharge rate modelled - grassland (%) | Recharge rate modelled - woodland (%) | Recharge rate modelled - combined (%) |
|---|---|---|---|---|---|---|---|---|
| 2010 | 266 | 356 | 317 | 355 | 33 | 44 | 39 | 44 |
| 2011 | 123 | 67 | 79 | 105 | 14 | 7 | 9 | 12 |
| 2012 | 351 | 152 | 204 | 203 | 35 | 15 | 20 | 20 |
| 2013 | 172 | 197 | 191 | 297 | 21 | 25 | 24 | 37 |
| 2014 | 202 | 701 | 441 | 698 | 28 | 98 | 62 | 98 |
| 2015 | 222 | 68 | 63 | 51 | 26 | 8 | 7 | 6 |
| 2016 | 129 | 51 | 49 | 48 | 18 | 7 | 7 | 7 |
| Average | 300 | 211 | 209 | 263 | 31 | 22 | 22 | 27 |

## 4 Discussion


### 4.1 Precipitation, soil moisture, and recharge events statistics

#### 4.1.1 Number of selected events

The fact that there are more precipitation than soil moisture events and more soil moisture events than recharge events is coherent with our expectations, as only parts of the precipitation events yield enough rainfall to cause a soil moisture response at 20 cm soil depth. Similarly, only a part of the water at 20 cm depths will contribute to recharge as parts are stored in the soil matrix or evaporated from the soil. However, another reason for precipitation events not resulting in a soil moisture event likely is due to the spatiotemporal heterogeneity of precipitation over our study area, heterogeneities induced

by throughfall in forest, and the distribution of soil profiles over the catchment. This probably distort the link between the precipitation signal and soil moisture or recharge.





From the total number of identified precipitation events, the portion of precipitation events leading to soil moisture events was slightly higher in woodland compared to grassland, and the portion of precipitation events that caused a soil moisture and a recharge response was higher in woodland compared to grassland. This is surprising as interception losses on forested
sites would suggest that less rainfall is infiltrating into the soil, and less is reaching the ground because of the water consumption by vegetation with deeper roots (Carriere et al., 2020). In another hand, Heilman et al. (2014) indicate that forest does not necessarily induce a bigger water consumption, especially with shallow soils with limited storage capacity. The denser root system in forest sites is likely an explanation for higher number of soil moisture events in woodland compared to grassland (Bargués-Tobella et al., 2014), with a higher soil heterogeneity in woodland and a higher hydraulic
conductivity (e.g. by macropores) through the soil. This explanation goes well with the observation that, in total, more soil moisture events in woodland sites caused a recharge response than on grassland sites. However, the threshold percentage of soil moisture leading to recharge is similar for both grassland and woodland.

The portion of precipitation events linked to recharge events but not to soil moisture events can likely be explained by the
limited number of soil moisture monitoring sites relative to the size of the catchment. But could also be explained by the exclusion of soil moisture events that were not linked to precipitation during the method's attribution step. In that case, the problem might be due to the cross-correlogram analyses that we used to estimate the attribution period between precipitation and soil moisture, which was applied and validated by Delbart et al., (2014) only to the estimation of the water transit time between precipitation and recharge, and not for soil moisture.


### 4.1.2 Soil moisture and recharge correlation

A range of soil moisture metrics was used for the correlation analysis: the antecedent conditions, the maximum reached value, the average, and the response amplitude of each event. Acceptable correlations were found between all soil moisture
metrics and recharge, except for the amplitude of the soil response. As shown in Fu et al. (2015), the soil wetting conditions before the recharge event have an important influence on karst recharge processes. This might explain why the soil response amplitude did not present a good correlation with effective infiltration: the antecedent conditions would be more important. The best and most consistent correlation among grassland, forest and their combination was found for the average moisture during the wetting period. This metric comprises the soil moisture conditions from the start of response, to the peak, and thus
better characterizes the soil moisture response. It turned out to be a better metric than the average soil moisture calculated for the entire event, which is typically biased towards the conditions during the long recession of the event. Its correlation results together with its above-mentioned characteristics made the average soil moisture during the wetting period the best metric to characterize recharge in this study.

The mean soil moisture during wetting plotted against the recharge volume showed an exponential relationship. A soil





moisture threshold seemed to have to be reached to activate the recharge; it was around 35 % (+/-8) of volumetric water content. The concept of such thresholds is common in karst modelling (Baker et al., 2020; Chen et al. 2017). Common soils over carbonate rock present porosities between 35 % and 65 % (Blume et al., 2010; Kirn et al., 2017). As the matrix potential is getting lower while soil moisture and the unsaturated hydraulic conductivity increase to initiate gravity driven

processes, we can make the approximation that these soils are getting close to saturation when porosities of 35-65 % are reached, to initiate percolation (Saxton et al., 1986). In our case, this threshold was between 35 % and 45 % of the volumetric water content, so this result was coherent with the type of soil at the studied catchment (silty clay). The standard deviation values of the soil moisture measurements over the catchment decreased during periods of high soil moisture and these were the periods when recharge was observed. This link was however stronger at the grassland sites compared to the

woodland sites. This was consistent with the fact that the soil moisture measurements in forests are more heterogeneous, in particular because of the trees, their distribution, their roots, and their transpiration (see above).

**4.2 Reliability of the event-based selection**

As discussed above, the total number of precipitation, soil moisture and recharge events selected with this approach is coherent with percolation, infiltration and evaporation processes.

The results of our study are based on a reliable extraction of rainfall events, soil moisture events and recharge events. The necessary thresholds were chosen according to previous studies and refined by different tests. The precipitation event

selection threshold was set to 1mm for each event, as in the approach used in Demand et al. (2019), inspired by Graham and Lin (2011) and Wiekenkamp et al. (2016). To end a precipitation event, a second threshold of 24h without precipitation was applied. This one was chosen after testing thresholds of 6h, 12h, 24h, and 48h. In the case of our study, the 24h was the threshold presenting the most coherent results: a smaller period was creating too many precipitation events linked to individual soil moisture reaction, while a bigger period was selecting nonrealistic precipitation events. The soil moisture

event selection threshold was set 1 % of the previous volumetric water content measurement, which corresponds to the accuracy of the probes. In a second step, soil moisture event attribution to precipitation event allowed to find the selected soil moisture events but not linked to precipitation. Those events were removed, based on the fact that no soil moisture response could not occur without precipitation (Fig. 2). This step allowed to reduce errors that the event selection based only on threshold could create. This is especially true for the soil moisture event extraction which relies only on the accuracy of the

probes. The recharge event selection threshold relied on the three-day running average of the observed slope of the discharge time series. Other event selection criteria might have resulted on a different set of recharge events. However, the selected recharge events represented 90 % of the total discharge volume observed in the catchments and therefore the method seemed appropriate for the purpose of this study.






## 4.3 Modelling of recharge based on soil moisture

The use of the drainage model based on the unit gradient approach allowed to estimate the recharge fluxes from soil moisture measurements. The two calibration parameters $ks$ and $B$ could be linked to the properties of the soil. This feature theoretically allows for applying the approach to karst sites with similar soil properties. In our case the fitted model parameter $ks$ was in the range of saturated hydraulic conductivity values corresponding to the soil found on the experimental catchment (silty clay) (Saxton et al., 1986). However, $ks$ and $B$ are effective model parameters and therefore cannot necessarily be derived from soil physical analysis. Given the simple approach, it is more meant that $ks$ is expected to be higher/lower in sites where the saturated hydraulic conductivity is also higher/lower. And the pore size distribution index $B$ is expected to be smaller the wider is the range of pore sizes are (Cary and Hayden, 1973). In our case, the $B$ value was higher for the grassland data. When calibrating the model parameters using the entire dataset (grassland and woodland sites) the effective parameters were between those determined for grassland and woodland sites with more similarity to the woodland sites. Still, the calibrated values and even the $ks$ values for the 10 % best model fits were in the range of saturated hydraulic conductivity values common for silty clay soil. By considering the 15 % best model fits, the resulting $ks$ values overlapped with those values fitted to only the grassland sites or the woodland sites. This suggests that common values of $ks$ and $B$ can be used for a successful simulation of recharge when a differentiation between grassland and woodland sites is not possible (Fig. 6). Larger saturated hydraulic conductivities going along with smaller $B$ values < 1, which results in low recharge at low saturation and very high recharge at or close to saturation, is in accordance with successful model representations of the soil/epikarst in previous modelling studies. These models use simple overflow bucket model that simulate zero recharge when below saturation and large volumes of recharge when saturated (Fleury et al., 2007).

When applying the event-based approach to estimate the sum of recharge volume of all events of a year, we noticed considerably differences between simulated and observed recharge volumes from year to year. One possible reason could be that our model was calibrated using events for which a precipitation event could clearly be linked to only a soil moisture events and a recharge event. As Fig. 4a shows, only a small portion of events fulfilled this criterion. From that point of view, it is intuitive that the model did not perform particularly well. This was especially true for the year 2014 when two events were highly overestimated compared to the others. They had the particularity to be attributed to a long wetting period, which lead to big volumes of simulated recharge.

The evaluation results are however acceptable on average on the entire studied period, with 88 % of recharge volume simulated using the combined areas. In general, the model's results were better with the combined areas data. As the combined areas soil moisture time-series was the result of the average of the grassland and woodland data, we were expected results in between the ones from grassland and woodland. However, it seemed that the $ks$ and $B$ calibration was more





influenced by the woodland data. The woodland covering the catchment in a bigger proportion (58 %), this could explain its
influence on the combined areas data. Also, the soil moisture measurements profiles that presented more events per year on
average were located in woodland (Fig. 4a) and the proportion of precipitation events resulting in soil moisture or recharge
event was higher in woodland than in grassland sites (Fig. 4b). While there is certainly the potential to improve the model or
calibrate it to more complex events, the simulated recharge for individual events and the average recharge over a longer
period of several years are promising given the simplicity of the model and the uncertainty of the model calibration
procedure.

## 4.4 Transferability of the method

The methodology presented in this study was developed with the aim to be applicable also at other karstified sites. Our
calibration showed that fitted parameters were not very sensitive to the simulated recharge at the catchment scale. A
distinction between grassland and woodland within the catchment was not necessary to obtain reasonable recharge rates. The
method can be applied to a single soil moisture profile but should be repeated to various locations on a catchment for better
representativeness. Especially in forest areas where the soil heterogeneities are larger. One single soil moisture monitoring
site would be limited in terms of being representative of the conditions across an entire catchment. The number and
distribution of profiles to be installed would also depend on the variability of the soil over the catchment. The probes should
be installed in the deepest possible depth to avoid evapotranspiration effect. Also, the locations of the precipitation
measurements need to be considered with care. Radar-based precipitation data could be an option to test in further studies.

Soil moisture data are available at various karst systems (Berthelin et al., 2020; Dorigo et al., 2021). However, recharge data
is not always fully available for the calibration. The parameter uncertainty analyses showed that the $B$ parameter tended to
fall below 1 when the $ks$ parameter was getting high (up to 30 mm d$^{-1}$). Since karst areas usually presents a very high rocks
permeability (Worthington et al., 2016), it is valid to assume high vertical saturated hydraulic conductivity ($ks$ even larger
than 50 mm d$^{-1}$) when estimating catchment-scale groundwater recharge. Consequently, parameter $B$ would assume values <
1, resulting in the behavior we found for the woodland and combined areas (Fig. 6b, c). These $ks$ and $B$ values could be a
good first guess for the model parameters, with possible in-situ $ks$ measurements for refinement. In addition, even if recharge
time-series are not fully available, a few observations such as shorter time series of discharge measurements conducted at a
spring, or other proxies for recharge such as groundwater heads measurements, conductivity, or water drops in a cave, could
be used for evaluation.



## 5 Conclusion

A method to estimate karst recharge using soil moisture measurements was developed and tested at a karst system in Southwest Germany. Based on precipitation, soil moisture, and discharge measurements, the method allowed the extraction of triplet combinations of single rainfall-soil moisture-recharge events. These combinations were then used to calibrate a drainage model that allowed deriving recharge fluxes and subsequent recharge volumes from soil moisture measurements. The application of the method to the test site showed the dominant influence of soil moisture measurements conducted in

woodland areas. This is the land cover where the highest number of soil moisture events was found, as well as the highest percentage of precipitation events creating soil moisture and recharge events. However, the usage of a combined time series of woodland and grassland soil moisture measurements allowed the best estimation of catchment recharge using the maximum available data and variability. The soil moisture averaged during the wetting period of each event was found to be the best indicator for estimating recharge. The relationship between soil moisture and recharge is exponential, with a

threshold of about 35 % of volumetric water content to initiate substantial recharge. The applied calibrated model allowed a reliable recharge volume estimation at the event scale. Adding up the event-scale recharge and comparing it to long-term observations, the model yielded 88 % of the observed recharge volume. The model calibration based on discharge measurements and converted into recharge volume leads to soil saturated hydraulic conductivity values coherent with the type of soil found at the test site and in accordance with existing recharge modelling concepts for karst systems. This means

that the approach could be applicable to different karst sites presenting different conditions using soil type characterisation for a priori estimation of the model parameters, i.e. without discharge data for calibration. The event approach also allows a semi- quantitative comparison of recharge from different time periods, climates, or locations where soil moisture and precipitation time series are available. The soil moisture probes used in this study are capable of direct measurements at a high temporal resolution and for a long period. In the future, the approach should be tested at different karst sites to explore

the ranges of its applicability to different catchment sizes, with different climate conditions, and different vegetation covers and soils. Other technical aspects, such as the number of soil profiles for measurements, could be explored in order to reproduce the method with optimum efficiency.

### Data availability

Soil moisture and climate data was provided by the Biodiversity Exploratory research project (DFG Priority Programme

1374) – Core Project Instrumentation. Streamflow data were provided by the Environment Agency of the German state of Baden-Württemberg (LUBW).



## Author contribution

The paper was conceived by RB under the supervision of AH, MM, and MR. RB, AH, MR, DD, and TO developed the methodology. RB conducted the formal analysis, with software support from TO, DD, and MS. TO developed the methodology and conducted the analysis of the recharge events volume computation. All authors participated in the writing – review & editing processes.

## Competing interests

The authors declare that they have no conflict of interest.

## Acknowledgements

We thank the instrumentation and remote sensing team of Thomas Nauss from Philipps University of Marburg for providing soil temperature and moisture data. We thank the managers of the three Exploratories, Kirsten Reichel-Jung, Iris Steitz, and Sandra Weithmannand (Swabian Alb), Katrin Lorenzen and Juliane Vogt (Hainich) and Miriam Teuscher (Schorfheide), all former managers for their work in maintaining the plot and project infrastructure; Christiane Fischer for giving support through the central office, Andreas Ostrowski for managing the central data base, and Markus Fischer, Eduard Linsenmair, Dominik Hessenmöller, Daniel Prati, Ingo Schöning, François Buscot, Ernst-Detlef Schulze, Wolfgang W. Weisser and the late Elisabeth Kalko for their role in setting up the Biodiversity Exploratories  project. The work has been funded by the DFG Priority Program 1374 "Infrastructure Biodiversity-Exploratories. RB, TO, MR, MS and AH were supported by the Emmy-Noether-Programme of the German Research Foundation (DFG, Grant No. HA 8113/1-1). MM was supported by the project PID2019-111759RB-I00 funded by the Spanish Research Agency, and this work is a contribution to the Research Group RNM-308 of Junta de Andalucía.

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
