# Peer review of "Estimating karst groundwater recharge from soil moisture observations - A new method tested at the Swabian Alb, Southwest Germany"

_Hydrology and Earth System Sciences, 2022_

## Author Comment (AC2)

Comment on hess-2022-291

Anonymous Referee #2

Referee comment on "Estimating karst groundwater recharge from soil moisture observations – A new method tested at the Swabian Alb, Southwest Germany" by Romane Berthelin et al., Hydrol. Earth Syst. Sci. Discuss., https://doi.org/10.5194/hess-2022-291-RC2, 2022

This paper tries to link precipitation- soil moisture- recharge relationship in catchment area scale of karst aquifer environment Good results and discussions are valid and clearly described. The contents of the paper are good and with valuable information to share with related field. Just some minor points suggest to check before the publishing:

*Reply: We thank the reviewer for her/his constructive comments that will contribute to improving the manuscript. According to her/his comments, we will perform and clarify the following points.*

1. There are limited information about the data distribution. For example, 15 sites of soil moisture measurement and discharge hydrograph. It's quite importance to evaluate characteristic from time series of different observation. I like to suggest to authors to add some typical hydrograph or data plots in suitable time windows.

*Reply: A figure with the time series of the different data used will be added to the manuscript in section 2.3.*

2. Some challenges rises after author use three spatially-combined-averaged soil moisture time series: grassland (G), woodland (W) and all area (C). It's needs to have some assumption and simplify the sites specific and spatially heterogeneity. Specially the karst surface - groundwater system will be spatially continuity such as G-W-G or W-G-W... from upstream to downstream. Such assumption as the paper describe could be only on infiltration processes and assume homogeneous after that.

*Reply: The idea to use spatially combined-average time series is used to average out site-specific patterns that are likely not representative of the average catchment behaviour. We wanted to avoid basing the analysis on a single time series that might be an outlier in terms of catchment-average behaviour.*

*We chose to average the observations in the grassland and woodland categories, assuming that these different vegetation types would have diverse influences on infiltration processes. Here, the tree and plant species are the same for the different observation points in the grassland and woodland. Furthermore, figure 5 of the paper showed that the standard deviation values of the soil moisture measurements over the catchment decreased when recharge was observed. The heterogeneities of measurements over the catchment are thus reduced when recharge is occurring.*

*Concerning the percolation processes, we assumed that the water infiltrated to the unsaturated zone is recharged according to the assumptions from Hartmann et al., 2021; Hartmann et al., 2012; Worthington et al., 2016.*

*We will add some text to make clear and describe the assumption of homogeneity in the system. We will also add some text to make clear that we assume that rainwater infiltrates into the soil and then vertically percolates down into the epikarst and karstic system, with one or two citations to justify this assumption. Under this assumption, a flow path from grassland to woodland to grassland or vice versa (G-W-G or W-G-W) is not a dominant flow path.*

3. Precipitation- soil moisture- recharge relationship as Fig.2, the paper chose "temporal delays" with a simple temporal buffer. Actually, the time lag or responses lag could be evaluating from cross check between two time-series. Such quantitative linkages also needed to support the following analysis and discussion.

*Reply: Indeed, we applied a cross-correlation analysis between the precipitation, soil moisture and discharge time series to estimate the temporal delays. The method applied was adapted from Delbart et al. 2014, who used the sliding-windows cross-correlation method. This information is given in section 2.1.2 of the paper. We will add some text about the approach to clarify this point.*

4. I am concerning the precipitation- soil moisture- recharge relationship analysis is event base, and the referred hydraulic parameters also random walks or with discrete distribution. For the calculation in catchment scale or annual flow in close years, such representative hydraulic parameters should be stable. It's better to check the timevariation of the related parameter between different events.

*Reply: We will check the variability of ks throughout different periods.*

---

## Author Response (AR1)

**Comment on hess-2022-291**

**Michael Stewart (Referee)**

Referee comment on "Estimating karst groundwater recharge from soil moisture observations – A new method tested at the Swabian Alb, Southwest Germany" by Romane Berthelin et al., Hydrol. Earth Syst. Sci. Discuss., https://doi.org/10.5194/hess-2022-291-RC1, 2022

General Comments

This paper uses soil moisture measurements to estimate recharge to a karst groundwater system via a soil drainage model. The karst outflow is from a single spring, whose discharge is used for comparison with the estimated recharge. The model performed reasonably well for single rainfall events and simulated 88% of the long-term average annual recharge volume for a Swabian Alb catchment.

The research question is well within the scope of HESS and presents novel concepts leading to a new method of estimating recharge in a karst groundwater system. The conclusions reached are substantial, relating to the validation of the method. The methods and assumptions are valid and clearly described. The experimental results are extensive and amply sufficient to support the interpretations and conclusions. Description of the method is clear and would allow the recharge estimation method to be applied to other catchments with the required data. The authors give adequate credit to related work and clearly describe their own contribution. The title is good, and the abstract is concise and appears complete.

Presentation is well structured and clear, and the language is satisfactory – some technical corrections are made below. Math formulae appear to be correct. There do not appear to be any unnecessary parts of the paper. The number and quality of references is satisfactory.

*Reply: We thank the reviewer for his positive and valuable comments that contribute to improving the manuscript. We applied all recommendations as suggested by the technical comments. Please find below the answers to the specific comments.*

Specific Comments

The method is original and ingenious, and works relatively well for the catchment tested, which has eight years of hourly data on rainfall, soil moisture and spring discharge available for testing. There may be problems with application to different catchments because of lack of data. In addition, catchments with substantially different types of recharge such as recharge from sinkholes or from streams flowing into sinks in their beds may present problems with implementation of the method. Larger catchments with very varied catchment areas may also present problems.

*Reply: Indeed, the lack of data might be a problem. For this method, it is needed to have precipitation, soil moisture and discharge data available. From our experience, soil moisture probes are robust, reducing the risk of data lack if the maintenance can be done frequently.*

*The presence of recharge from sinkholes would create more recharge events linked to precipitation but not to a soil moisture event. This could be highlighted during the first steps of the method. However, the recharge volume estimation from soil moisture would be indeed reduced in that case.*

*We specified it in the manuscript in the discussion section.*

There are considerable assumptions/requirements with the method. 1. The catchment area must be delineated accurately, this may be difficult in some areas. 2. Contributions from different vegetation covers and soils (as in this study) need to be assessed by multiple soil moisture measurement sites. 3. For comparison with the recharge estimated from the soil measurements, the spring discharge should be able to accurately represent groundwater recharge. This may be difficult in systems with several outlets.

*Reply: The catchment area and discharge data that represents accurately recharge need to be defined to estimate the volume of recharge. Nevertheless, if this is not possible, the method still can be used to predict the recharge occurrence and relative dynamics, with other recharge measurements such as drip rates from a cave, discharge from epikarst outlet etc..*

*We added these points to the manuscript in the discussion.*

**Comment on hess-2022-291**

**Anonymous Referee #2**

Referee comment on "Estimating karst groundwater recharge from soil moisture observations – A new method tested at the Swabian Alb, Southwest Germany" by Romane Berthelin et al., Hydrol. Earth Syst. Sci. Discuss., https://doi.org/10.5194/hess-2022-291-RC2, 2022

This paper tries to link precipitation- soil moisture- recharge relationship in catchment area scale of karst aquifer environment Good results and discussions are valid and clearly described. The contents of the paper are good and with valuable information to share with related field. Just some minor points suggest to check before the publishing:

*Reply: We thank the reviewer for her/his constructive comments that contribute to improving the manuscript. According to her/his comments, we performed and clarify the following points.*

1. There are limited information about the data distribution. For example, 15 sites of soil moisture measurement and discharge hydrograph. It's quite importance to evaluate characteristic from time series of different observation. I like to suggest to authors to add some typical hydrograph or data plots in suitable time windows.

*Reply: A figure (Fig 3) with the time series of precipitation, soil moisture in grassland as an example and discharge, was added to the manuscript in section 2.3.*

2. Some challenges rises after author use three spatially-combined-averaged soil moisture time series: grassland (G), woodland (W) and all area (C). It's needs to have some assumption and simplify the sites specific and spatially heterogeneity. Specially the karst surface - groundwater system will be spatially continuity such as G-W-G or W-G-W... from upstream to downstream. Such assumption as the paper describe could be only on infiltration processes and assume homogeneous after that.

*Reply: The idea to use spatially combined-average time series is used to average out site-specific patterns that are likely not representative of the average catchment behaviour. We wanted to avoid basing the analysis on a single time series that might be an outlier in terms of catchment-average behaviour.*

*We chose to average the observations in the grassland and woodland categories, assuming that these different vegetation types would have diverse influences on infiltration processes. Here, the tree and plant species are the same for the different observation points in the grassland and woodland. Furthermore, figure 5 of the paper showed that the standard deviation values of the soil moisture measurements over the catchment decreased when recharge was observed. The heterogeneities of measurements over the catchment are thus reduced when recharge is occurring.*

*Concerning the percolation processes, we assumed that the water infiltrated to the unsaturated zone is recharged according to the assumptions from Hartmann et al., 2021; Hartmann et al., 2012; Worthington et al., 2016.*

*We added some text to make clear and describe our assumption about infiltration, percolation and recharge processes in section 2.3 with additional citations.*

3. Precipitation- soil moisture- recharge relationship as Fig.2, the paper chose "temporal delays" with a simple temporal buffer. Actually, the time lag or responses lag could be evaluating from cross check between two time-series. Such quantitative linkages also needed to support the following analysis and discussion.

*Reply: Indeed, we applied a cross-correlation analysis between the precipitation, soil moisture and discharge time series to estimate the temporal delays. The method applied was adapted from Delbart et al. 2014, who used the sliding-windows cross-correlation method. This information is given in section 2.1.2 of the paper. We added some text about the approach to clarify this point.*

4. I am concerning the precipitation- soil moisture- recharge relationship analysis is event base, and the referred hydraulic parameters also random walks or with discrete distribution. For the calculation in catchment scale or annual flow in close years, such representative hydraulic parameters should be stable. It's better to check the timevariation of the related parameter between different events.

*Reply: We checked the variability of ks throughout different periods. Our dataset was divided into three parts. Each sub-dataset was used to calibrate the model. Except for the first third of the data set, which includes fewer large recharge events, the two other parts led to similar simulations to the one introduced in the paper.*

| | Grassland | | | Woodland | | | Combined areas | | |
|---|---|---|---|---|---|---|---|---|---|
| Period | 1/3 | 2/3 | 3/3 | 1/3 | 2/3 | 3/3 | 1/3 | 2/3 | 3/3 |
| ks (mm d$^{-1}$) | 48,4 | 15,9 | 12 | 7,8 | 50 | 49,2 | 9,8 | 48,2 | 31,9 |
| B (-) | 0,4 | 0,7 | 5 | 5 | 0,5 | 0,7 | 5 | 0,3 | 0,95 |

**Remarks from the preceding review file validation**

Your tables contain coloured cells or/and coloured values. Please note that this will not be possible in the final revised version of the paper due to HTML conversion of the paper. When revising the final version, you can use footnotes or italic/bold font. But if the colour spectrum is necessary and cannot be exchanged for footnotes, bold, or italic, then please inform us via email. 2. Please ensure that the colour schemes used in your maps and charts allow readers with colour vision deficiencies to correctly interpret your findings. Please check your figures using the Coblis – Color Blindness Simulator (https://www.color-blindness.com/coblis-color-blindness-simulator/) and revise the colour schemes accordingly.

*The coloured cells from table 2 were removed. Colours from figures 6 and 7 were modified to allow readers with colour vision deficiencies to correctly interpret our findings.*